# Non-Interventional and High-Precision Temperature Measurement Biochips for Long-Term Monitoring the Temperature Fluctuations of Individual Cells

**DOI:** 10.3390/bios11110454

**Published:** 2021-11-15

**Authors:** Danhong Han, Jingjing Xu, Han Wang, Zhenhai Wang, Nana Yang, Fan Yang, Qundong Shen, Shengyong Xu

**Affiliations:** 1Key Laboratory for the Physics & Chemistry of Nanodevices, Department of Electronics, Peking University, Beijing 100871, China; handanhong@126.com (D.H.); wzh632@163.com (Z.W.); yangnana@pku.edu.cn (N.Y.); fyang1992@pku.edu.cn (F.Y.); 2Beijing Research Institute of Mechanical Equipment, Beijing 100854, China; 3School of Microelectronics, Shandong University, Jinan 250100, China; 4Shenzhen Research Institute, Shandong University, Shenzhen 518057, China; 5Department of Orthopedics, Air Force Medical Center, Beijing 100142, China; wh.23@hotmail.com; 6Department of Chemistry, Nanjing University, Nanjing 210023, China; qdshen@nju.edu.cn

**Keywords:** single-cell temperature measurement, temperature sensors, non-interventional biosensors, biochip fabrication, cell temperature, cell metabolism

## Abstract

Monitoring the thermal responses of individual cells to external stimuli is essential for studies of cell metabolism, organelle function, and drug screening. Fluorescent temperature probes are usually employed to measure the temperatures of individual cells; however, they have some unavoidable problems, such as, poor stability caused by their sensitivity to the chemical composition of the solution and the limitation in their measurement time due to the short fluorescence lifetime. Here, we demonstrate a stable, non-interventional, and high-precision temperature-measurement chip that can monitor the temperature fluctuations of individual cells subject to external stimuli and over a normal cell life cycle as long as several days. To improve the temperature resolution, we designed temperature sensors made of Pd–Cr thin-film thermocouples, a freestanding Si_3_N_4_ platform, and a dual-temperature control system. Our experimental results confirm the feasibility of using this cellular temperature-measurement chip to detect local temperature fluctuations of individual cells that are 0.3–1.5 K higher than the ambient temperature for HeLa cells in different proliferation cycles. In the future, we plan to integrate this chip with other single-cell technologies and apply it to research related to cellular heat-stress response.

## 1. Introduction

The temperature of a single cell changes with its metabolic state and in response to external stimuli [1,2,3]. Accurate real-time temperature measurements of individual cells in different phases of their life cycles and in different external environments is important for understanding the internal connection between metabolism and cell function [4], as well as to promote research on organelle function [5], heat treatment of cancer [6], and drug screening [7].

Semi-contact, temperature-sensitive fluorescence probes are widely designed to measure the cellular temperature distribution directly, in contradistinction to measurements of the fluorescence lifetime and intensity and the spectral-peak shift and anisotropy of fluorescent materials, including fluorescent polymers, lanthanide nanoparticles, quantum dots, nanodiamonds, and molecular beacons [8,9]. Fluorescent probes have the advantages of spatial resolution as high as hundreds of nanometers and thermal resolution of 0.01–1 K [10,11], but they are limited in accuracy and stability because the fluorescent materials are easily affected by the complex intracellular environment, which includes different ion concentrations, cytoplasmic viscosity, PH, and other factors [12,13]. However, the introduction of fluorescent materials also may affect the normal metabolic activities of cells. For instance, using different fluorescent probes, very different temperatures have been reported for HeLa cells, ranging from 1 K to 10 K above the ambient temperature [14,15,16,17,18].

A thermocouple sensor detects the temperature of the object with which it is in contact more accurately by collecting heat directly from the contact interface between the object and the sensor. Thermocouple sensors obtain the absolute temperature of an object according to the Zeroth Law of Thermodynamics [19,20]. Ning Gu et al. developed a sandwich-structured Pt-Wu thermocouple probe to measure the temperature response of human glioma cells (U251) to different chemical stimuli [21]. In 2019, Sanjiv’s team at the University of Illinois prepared a miniature thermocouple probe on a silicon nitride (Si_3_N_4_) tip to measure the transient heat production during the uncoupling of mitochondrial protons in the neurons of California sea hares, and they detected a temperature rise of 7.5 K [22]. However, the temperature spike was not stable, and it decayed at the rapid rate of 4.8 K/s. In addition to invasive thermocouple probes, thin-film thermocouple (TFTC) sensors have also been used to measure the absolute temperatures of adhering cells. Yang et al. fabricated Pt–Cr thermocouples on a glass substrate to monitor the temperature change of human-liver cancer cells, although the accuracy of these results has been questioned [23]. Hence, a new structural-design concept is required to obtain accurate, non-interventional, and stable temperature measurements for individual cells.

In this article, we report the design and fabrication of a new cellular temperature-detection biochip, introducing Pt–Cr thermocouples array, a freestanding Si_3_N_4_ platform and a dual-temperature control system to reduce thermal noise and to improve temperature sensitivity. We used this chip to measure the temperature fluctuations of individual HeLa cells quantitatively both in their basal metabolic state and when subject to HCl stimuli. These measurements have verified the feasibility of the chip for use in accurate, non-interventional, and real-time temperature monitoring of individual cells.

## 2. Materials and Methods

### 2.1. Chip Fabrication

We fabricated the chips using standard micro-nano processing techniques [24] on 4-inch Si <100> wafers. The wafers were coated on both sides with 400-nm-thick Si_3_N_4_ by using plasma-enhanced chemical-vapor deposition (PECVD) at the Microsystem Institute, Chinese Academy of Sciences, Shanghai, China. We performed the photolithography processes on a MJB4 mask aligner (SUSS MicroTec, Garching, Germany), and we removed the residual photoresist on the patterned substrates by utilizing an oxygen from a plasma generator (PDC-M, PVA TePla, Wettenberg, Germany) plasma at a power of 250 W for 30 s. We deposited the Pd and Cr thin films successively with an electron-beam evaporator (DE400, DE Tec, Beijing, China) and a magnetron sputtering system (PVD75, Kurt J. Lesker, Pittsburgh, PA, USA) operating in Ar atmosphere, respectively.

We implemented detailed wet-etching procedures to fabricate the freestanding Si_3_N_4_ windows using published methods [25]. In brief, we employed a unique technique developed by Wang et al. to complete the double-side alignment with a single-side mask aligner (MJB4) after the TFTC arrays were fabricated [26]. We patterned the open windows for wet etching on the backsides of the substrates and etched them with an inductively coupled plasma system (ICP, Technology Minilock III, TRION). Then we etched the substrate backside in 30 vol.% KOH at 80 °C–85 °C using a flotation technique. We created concave, pyramid-like patterns in the backside window in 7–8 h by taking advantage of the high ratio of the etching rate for Si <100> over Si <111>, until penetrating square windows were seen from the front side. Finally, we deposited a 5-nm-thick HfO_2_ thin film on the device surface with an atomic-layer-deposition apparatus (ALD, Savannah S100, Cambridge NanoTech, Cambridge, MA, USA) to provide electrical insulation on the thermocouple surface. An optical micrograph of unit A on the device is shown in Figure 1b. We used polydimethylsiloxane (PDMS) and centrifuge tubes to create a cell-culture pool with an inner diameter of 10 mm and a height of 35 mm in the center of the chip.

### 2.2. Molecular Modification of the Surfaces of the Chips

The molecular-modification steps we used are as follows: (1) We dropped 400 μL of poly-l-lysine solution into the cell-culture pool on the chip after it had been sterilized; (2) We aspirated the poly-l-lysine solution 20 min later; (3) We rinsed the surface of the cell-culture pool slowly with phosphate buffered saline (PBS) to complete the modification with poly-l-lysine.

### 2.3. Cell Culture and Temperature Monitoring

After resuscitation, we cultured the HeLa cells in a DMEM F12 cell-culture medium containing 10% fetal bovine serum and 1% double antibody in the 37 °C incubator with a 5% carbon dioxide mixed atmosphere. The culture medium was replaced every two days, and cells were used in 90% confluence. All the cells used in the temperature-measurement experiments were from the 3rd–6th generation after resuscitation. We placed the suspended cells—with a seeding density of about 30%, mixed with 2.5 mL of cell-culture solution—into the culture pool of the chip, and we then placed the chip into the incubator and subsequently started the measurement system. The output-temperature signals from the TFTCs usually need dozens of minutes to achieve stability after the system is started.

## 3. Results and Discussions

### 3.1. Design and Properties of the New Biochip

#### 3.1.1. The Temperature-Measurement Chip and Control System

The functional region for cellular temperature detection in the newly designed chip consists of a temperature sensor array on a freestanding Si_3_N_4_ thin film. Figure 1a illustrates the cross-sectional structure of this chip graphically. Figure 1b is a photograph of the finished chip, in which three independent units—named A, B, and C—are designed with the same sensor layout but with different substrates. The original sandwich structure of Si_3_N_4_ (400 nm)–Si (410 μm)–Si_3_N_4_ (400 nm) is retained for unit B, while the Si_3_N_4_ layer on the back and the middle Si layer are etched away for units A and C to leave the freestanding Si_3_N_4_ thin film. As a result, units A and C serve as the experimental groups, and unit B is the control group in this study. Each unit is fabricated with 18 sensors of Pd–Cr TFTCs, each with a sensor size of 2 × 2 μm^2^. A snake-shaped heater in the center of the chip is designed to modify the ambient temperature around the cells for a future study.

Integrated with the chip is a home-made cell-temperature measurement system we developed that consists of three components: a data-processing module, a dual-temperature control setting, and an in-situ drug-delivery device. The data-processing module uses a nanovoltmeter (Keithley 2182A), a 10 × 10 multiplexer, and a LabVIEW program to control the data-collection cycle (0.08–1 min). The dual-temperature control setting holds the temperature to be 37 °C in the cell incubator (LiKang HF240) that houses the chip, and it maintains a temperature-controlled box outside the cell incubator, in which the nanovoltmeter and multiplexer are placed, at the set temperature of 27 ± 0.1 °C. In this way, we are able to reduce measurement errors caused by temperature drift in the cold ends of TFTCs and by temperature differences between the instruments. An in-situ drug-delivery device that can be controlled externally is placed within the cell incubator to reduce the temperature interference resulting from the operation of adding drugs. These aspects of the system design reduce the thermal noise and help to stabilize the measurement environment. Figure 1c shows the settings and illustrates the operation of the whole system.

#### 3.1.2. Improving Cell Compatibility by Molecular Modification of the Chip Surface

The key to successful contact-temperature measurements with TFTCs is to make cells attach to the temperature sensors to form a good thermal interface. However, we found that it was difficult for cultured HeLa cells to adhere well to the chip. To solve this problem, we explored the effects on cell adhesion of molecular modifications of the surfaces of the chips with poly-l-lysine and polydopamine.

We modified two chips, one with poly-l-lysine and one with polydopamine, and used the third, unmodified chip as the control group. We seeded a suspension of HeLa cells with a density of about 30% onto each of the three chips and then placed them into the system for culturing and temperature monitoring. Photos of the 48- and 72-h morphologies of the HeLa cells, together with their temperature changes on the different chips, are shown in Figure 2. Figure 2a shows that, on the chips modified with poly-l-lysine and polydopamine, the spindle-shaped HeLa cells cover the entire surface of the chip tightly, and they grow better than those on the unmodified chip, on which it is difficult to get the HeLa cells to grow and adhere. Compared with poly-l-lysine-modified chips, for the polydopamine-modified chips it is difficult to detect the temperature-fluctuation signals from the cells, as shown in Figure 2b, because the polydopamine film is too thick to weaken the heat transfer from the cells to the thermocouple sensors. Meanwhile, the temperature data curve of HeLa cells in the control chip has a lower fluctuation. Therefore, we chose poly-l-lysine to modify the chips’ compatibility with the cells in this study to ensure that the under-test HeLa cells can grow on and adhere to the chip better, enabling accurate temperature measurements.

#### 3.1.3. High Thermal Sensitivity

The capacity of the substrate to dissipate heat affects the transient thermal equilibrium of the interface, which in turn affects the temperature sensitivity and resolution of the thermocouple sensors. We monitored the temperatures of HeLa cells on the same chip but on different substrates—the freestanding Si_3_N_4_ substrate and the solid Si_3_N_4_/Si/Si_3_N_4_ substrate—to explore the effect of introducing the freestanding Si_3_N_4_ platform on the thermal sensitivity of the chips.

The curves labeled 2Au2, 2Bu8, and 2Cd3 in Figure 3a—named for the corresponding TFTC numbers—are a set of typical thermoelectric signals from the three units of a blank control chip exposed only to the culture medium (without HeLa cells). They show a background temperature ranging from about −30 to −20 mK. In comparison, the curves labeled 1Au2, 1Au7, 1Cu8, and 1Cd2 in Figure 3b come from HeLa cells on the freestanding Si_3_N_4_ substrate. They show an obvious fluctuation range of ~33 μV—or 1400 mK—and an average temperature higher than the ambient temperature by 900 mK. The variation of fluctuations of 1Au2, 1Au7, 1Cu8 and 1Cd2 comes from the different metabolism of the in-test cells attached to them. This confirms that the chip can indeed collect temperature data from the under-test cells.

In contrast, the thermoelectric signals labeled 1Bu3, 1Bu7, and 1Bd5 in Figure 3b,c—all collected from HeLa cells on the solid Si_3_N_4_/Si/Si_3_N_4_ substrate—do not show any obvious fluctuations, and they almost coincide with the background-temperature curves from 2Au2, 2Bu8, and 2Cd3. This occurs because the thermal conductivity of the hundreds-of-nanometers-thick Si_3_N_4_ film was measured to range from 0.5 to 10 W·m^−1^·K^−1^ [27,28], which is much lower than that of Si (148 W·m^−1^·K^−1^) [29]. Moreover, the thermal conductivity of the Si layer below the Si_3_N_4_ film in the Si_3_N_4_/Si/Si_3_N_4_ substrates is much higher than the value of 0.0256 W·m^−1^·K^−1^ for the freestanding substrate in air [30]. Thus, the introduction of the freestanding Si_3_N_4_ platform greatly reduces heat diffusion and thus enhances the thermal sensitivity of the TFTC sensors, which improves the accuracy of the temperature measurements.

#### 3.1.4. Low Thermal Noise

We further tested both the thermal noise of the freestanding TFTC sensors on the chips in the home-made temperature-measurement system and the thermal noise from other standard commercial thermocouples. The commercial thermocouples are made of two materials with different Seebeck coefficients by use of fusion welding. For example, K-type thermocouples are made of a nichrome and nickel-silicon alloy. It has the diameter of generally 1.2–4.0 mm, a good stability and a long service life, but is not suitable for contact measurement of the surface and internal temperature of tiny objects like cells. The developed freestanding TFTC sensors here are prepared by a thin film deposition process, and thus have smaller sizes within the range of hundreds of nanometers to dozens of micron meters. In contrast, they should have a high sensitivity and a lower thermal noise due to the microscopic electronic transport.

We inserted the measuring ends of three standard commercial thermocouples—types J, T, and K—totally into a pure culture medium without HeLa cells on the developed chip. The 24-h thermoelectric and temperature outputs of the four kinds of thermocouples are shown in Figure 3d,e. The average fluctuations of the output voltages from the J-, T-, and K-type commercial thermocouples are all 2–3 μV, implying background-temperature fluctuations of 50–60 mK according to their calibrated temperature-sensing coefficients. In contrast, the voltage curves collected from our developed chip ranged within only 0.2 μV, corresponding to background-temperature fluctuations of 10 mK, using the temperature-sensing coefficient of 22.36 ± 1.0 μV·K^−1^ for Pd−Cr thermocouples on freestanding Si_3_N_4_ platforms [25], with the calibration method reported [23]. Compared with standard commercial thermocouples, the thermal noise from the developed chips is reduced by a factor of five.

### 3.2. Preliminary Applications in Temperature-Measurements of Individual Cells

#### 3.2.1. Temperature Fluctuations from the Basic Metabolism of Hela Cells

Using the developed chips, we monitored the temperature fluctuations due to the basic metabolism of the adhering HeLa cells for 48 h. We performed no unboxing operations during the whole measurement process.

Figure 4a–d show four typical curves of the changes in absolute temperature above the ambient temperature (37 °C) during a timespan of 8–48 h from the four HeLa cells on the thermocouples named 1Au2, 1Au7, Cu8, and 1Cd2. They show that the temperatures of the four HeLa cells have similarly large fluctuation ranges but with different trends. Specifically, the absolute thermoelectric potentials from sensors 1Au2, 1Au7, 1Cu8, and 1Cd2 range within 0–25.6 μV, 0–34.1 μV, 0–25.5 μV, and 0–29.5 μV, respectively. Using their thermoelectric conversion coefficient of 22.36 μV·K^−1^, we obtain the corresponding range of temperature variations for the four cells to be 350–1520 mK, indicating that the temperatures of HeLa cells in normal life should be about 0.35–1.52 K higher than the ambient temperature. In addition, the temperatures measured by the four thermocouples at the 48 h point were 685 mK, 705 mK, 1150 mK, and 1040 mK, respectively, consistent with the positions of the HeLa cells at 48 h. As shown in the optical-microscope images taken at 48 h, Figure 4e,f, HeLa cells are attached to thermocouples 1Au2, 1Au7, 1Cu8, and 1Cd2, as shown by the red arrows. Among these, the measuring ends of 1Au2 and 1Au7 are far from the nuclei of the corresponding HeLa cells, while the ends of 1Cu8 and 1Cd2 are located in the core areas of the HeLa cells. The nucleus and mitochondria located in the cellular core area are the main heat sources in normal cell metabolism [31,32,33], and that is why the cell temperatures collected from 1Cu8 and 1Cd2 are higher than those from 1Au2 and 1Au7. The temperature fluctuations of HeLa cells—measured here using the developed chips—have been widely reported, mostly from the use of semi-contact fluorescence probes. As early as 2010, Suzuki’s team at Waseda University used rare-earth metal complexes to demonstrate that the temperatures of HeLa cells are higher than the ambient temperatures by about 1 K [34]. Takeharu et al. reported that the temperature difference between the cytoplasm and the nucleus can be as much as 2.9 K. Later, the temperature of mitochondria was found to increase by 6–9 K after being stimulated by molecular-messenger fluorescent thermosensitive dyes [16].

Another important finding is that the temperature change collected by sensor 1Au2 in Figure 4a has very obvious peaks and troughs. The time interval between the two peaks marked by vertical red dashed lines is about 1250 min, i.e., 21 h, which is very close to the HeLa proliferation cycle of 22–24 h [35,36]. We therefore surmise that there is a cell-reproduction cycle between these two peaks and that the peaks in this temperature curve correspond to the most vigorous metabolic-activity stages of the cell-growth cycle, such as phases G1 [37] or G2 [38]. The cell-temperature change collected by sensor 1Au7 in Figure 4b also seems to have such a double-peaked temperature fluctuation. However, the temperature curves measured simultaneously by sensors 1Cu8 and 1Cd2 in the same chip do not show any such fluctuations. This may result from the migration of the HeLa cells across the chip surface during the measurement process, as has been reported previously [34,39], so that the contact site between the cell and the sensor end is not fixed. Thus, when the cellular core area happens to be attached to the measuring end, the sensor receives a larger thermoelectric signal, and when its core area leaves the end, the output-temperature signal drops.

#### 3.2.2. Temperature Drop of Hela Cells Subjected to a Lethal Fluid

To explore the feasibility of applying this temperature-measurement chip in cell heat-stress research, we monitored the temperature response of HeLa cells in real time during in-situ application of a lethal fluid. As the lethal fluid, we slowly added 400 μL of concentrated hydrochloric acid (HCl, 12 mol/L) at the same temperature as the culture medium dropwise to the culture pool containing the HeLa cells. At the same time, we added concentrated HCl with the same concentration, volume, and temperature to a culture pool without cells as a blank control group.

Figure 5a shows the output thermoelectric signals from two HeLa cells on thermocouples 3Ad7 and 3Au8. Before injection, temperature fluctuations 100–1500 mK above the ambient temperature were collected by these two sensors. After injecting the concentrated HCl, both temperature signals dropped abruptly to the ambient temperature and remained there for the remaining monitoring time of several hours. In contrast, a transient temperature spike of about 300 mK was produced by the HeLa cells on the Si_3_N_4_/Si/Si_3_N_4_ substrate when subjected to concentrated HCl, as illustrated in Figure 5b. Once again, this proves that the introduction of the freestanding Si_3_N_4_ platform improves the temperature sensitivity and accuracy of the temperature-measurement chips.

We performed the same lethal experiments with a control group without an HeLa cell culture. We observed no steep drop in temperature in the control group, except for a transient temperature spike, as shown in Figure 5c. This confirms that the temperature drop shown in Figure 5a is not caused by the introduction of the concentrated HCl itself, but instead is caused by the death of the HeLa cells due to the addition of concentrated HCl. Furthermore, the experimental finding of exothermic heat release upon the addition of concentrated acid, shown in Figure 5d, confirms that the introduction of concentrated HCl causes the temperature of the culture medium to rise, which proves again that the temperature drop seen in Figure 5a was not caused by the concentrated HCl itself. Figure 5d also explains why we chose concentrated HCl as the lethal fluid instead of concentrated sulfuric acid (H_2_SO_4_) with a mass fraction of 98%: the exothermic effect of concentrated H_2_SO_4_ is much stronger than that of concentrated HCl. The morphological changes of the HeLa cells following exposure to concentrated HCl are shown in Figure 4e. All the cells are fragmented, and normal metabolic activities are terminated, within 2 min after the injection of concentrated HCl; this is the reason why the cell temperature drops so quickly in Figure 5a.

These experimental results exclude the possibility of false temperature signals caused by factors such as changes in the ionic solution, surface discharges on the cell membrane, or environmental thermal noise, providing strong evidence that the chips we developed here are reliable, accurate, and stable in detecting cellular temperature changes as small as a dozen millikelvins and that they have great potential for applications in cell heat-stress research.

Compared with semi-contact fluorescent thermosensitive probes, which use various fluorescent characteristics that are easily affected by the surrounding environment, the on-chip freestanding TFTC sensor chip we have developed measures the cellular temperature with a higher stability and is not easily affected by complex environments that include various ions, biochemical molecules, and a changing PH. Compared to the invasive thermocouple probes, the developed chip is non-intrusive and non-destructive. Furthermore, this chip has the advantages of high thermal sensitivity, high temperature resolution, low thermal noise, and the capability of monitoring the cell-temperature response to external stimuli for as long as several days. In addition, the chip can be further improved to study the temperature distribution of a single cell by equipping it with sub-micron scale temperature sensors and restricting walls. With the help of a live-cell dynamic monitoring system, information that includes the cellular morphology and relative position—as well as real-time temperature measurements by the sensors on the chips—can be obtained at the same time to determine the 2D temperature distribution of a single cell via a thermodynamic model of the cellular temperature distribution. This work thus provides a stable and reliable biochip for investigations of single-cell technology and cell heat-stress response.

## 4. Conclusions

We have demonstrated an accurate, non-interventional, high-precision and long-term temperature-measurement chip for individual cells. Combining the chip with a home-made control system, we found that individual HeLa cells adhering to the sensor exhibited temperature fluctuations 0.3–1.5 K higher than the ambient temperature in their basal metabolism and that this temperature excess decayed rapidly when subjected to in-situ HCl stimuli, which confirms the accuracy and feasibility of this single-cell temperature-measurement chip. This simple yet challenging experiment provides a new method of monitoring cellular temperatures that can be further integrated with other single-cell technologies to enable fundamental research such as in-situ and real-time temperature monitoring of cancer cells subject to different drugs.

## Figures and Tables

**Figure 1 biosensors-11-00454-f001:**
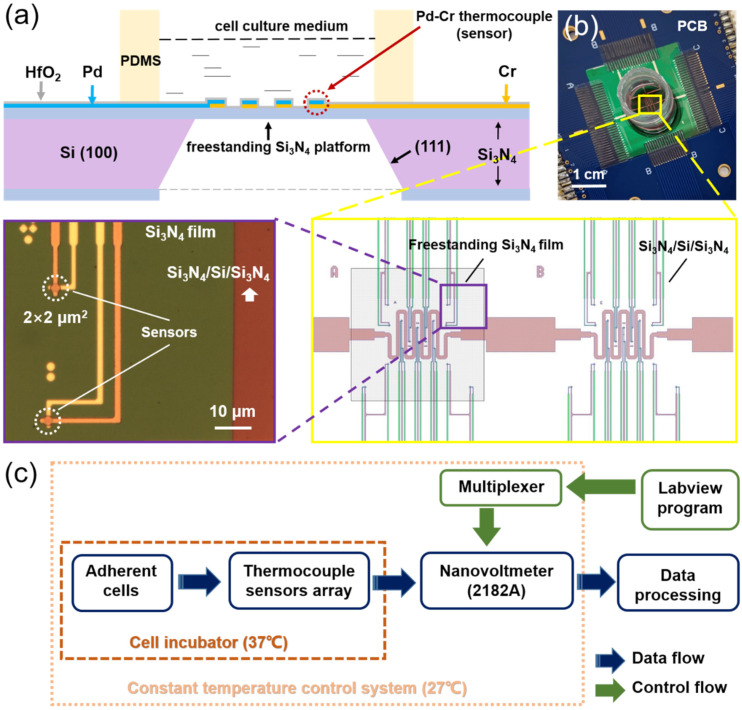
Schematic illustration of the new biochip and measurement system. (**a**) Schematic diagram of a cross-section of the functional area on the chip. The cream-colored bars labeled PDMS represent a cross-section of the polydimethylsiloxane ring that contains the cell-culture pool (see Experimental Section). (**b**) Optical photo of the whole chip, including the functional area. The chip is mounted on a printed-circuit board (PCB). (**c**) The working principle of the homemade, low-noise cell-temperature detection system.

**Figure 2 biosensors-11-00454-f002:**
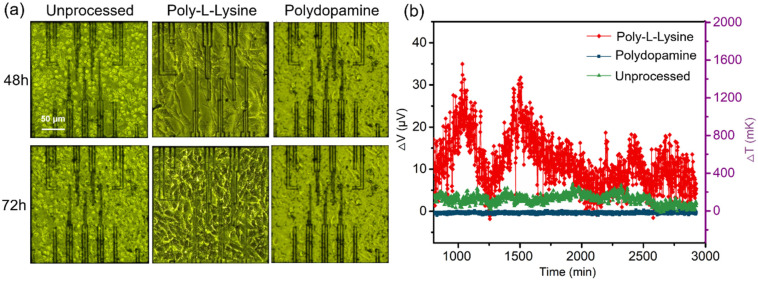
Effects of surface modifications of the chips. (**a**) Photos of the 48- and 72-h morphologies of HeLa cells on chips with different surface treatments. (**b**) Cell-temperature data collected from the unprocessed chip and from chips with poly-l-lysine and polydopamine modifications.

**Figure 3 biosensors-11-00454-f003:**
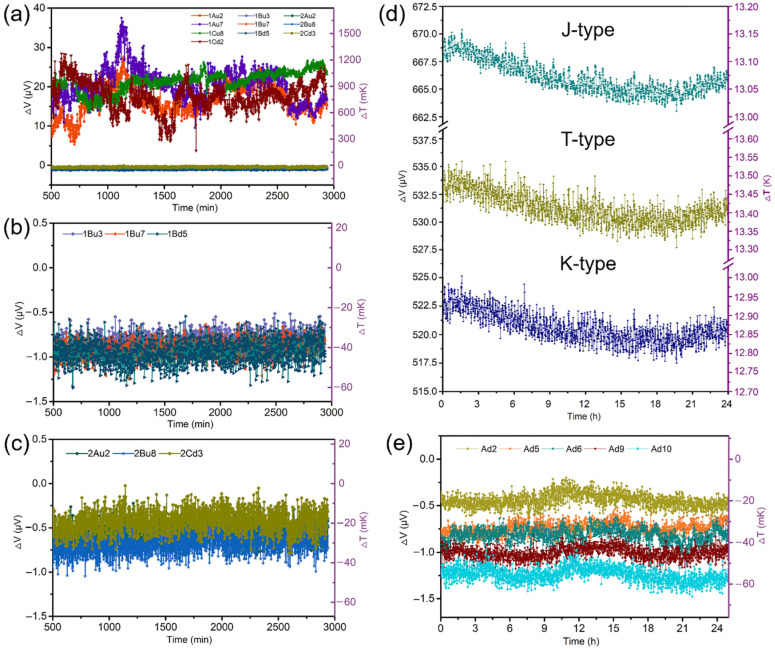
The high thermal sensitivity and low thermal noise of the cellular temperature-monitoring chip and system. (**a**) Background-temperature curves from the developed chips. (**b**) Comparing the output thermoelectric signals from the temperature sensors on freestanding Si_3_N_4_ platform and on solid Si_3_N_4_/Si/Si_3_N_4_ substrate exposed to a HeLa cell culture and to a blank control group without cells shows the high thermal sensitivity of the chips. (**c**) Enlarged temperature curves of HeLa cells obtained from the sensors on the Si_3_N_4_/Si/Si_3_N_4_ substrates show a low temperature sensitivity. (**d**) Thermal-noise test results for three standard commercial thermocouples. (**e**) Thermal-noise test results for the developed chips with Pd–Cr TFTC sensors on a freestanding Si_3_N_4_ platform.

**Figure 4 biosensors-11-00454-f004:**
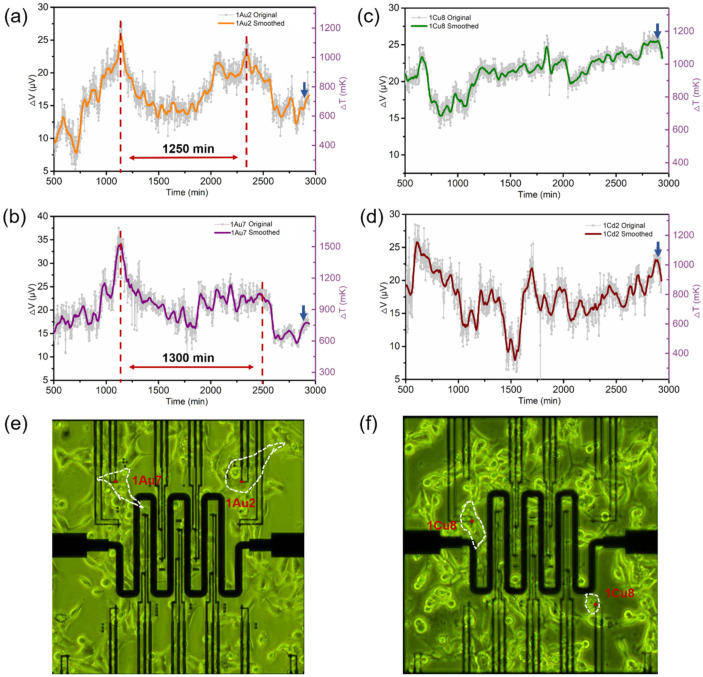
(**a**–**d**) The absolute temperature fluctuations of the HeLa cells on the four TFTC sensors named 1Au2, 1Au7, 1Cu8, and 1Cd2 in a 37 °C environment after removal of the background thermal noise. The gray curves represent the original measurement data, and the orange, purple, green, and dark red thick curves result from smoothing the original data. (**e**,**f**) Photos of the relative positions of the under-test cells and the four thermocouple sensors are highlighted by red dots.

**Figure 5 biosensors-11-00454-f005:**
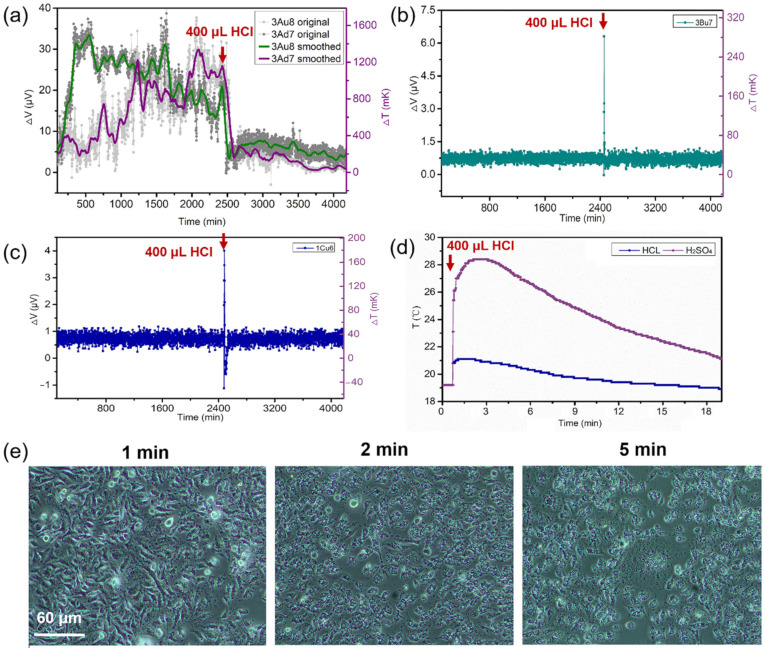
The temperatures of the HeLa cells drop sharply upon the addition of a lethal fluid (the red arrows represents the moment when concentrated HCl is injected into the medium). (**a**) Two typical curves showing the temperature changes caused by the deaths of the HeLa cells due to the addition of concentrated HCl. (**b**) The control group shows no obvious temperature drop in the HeLa cells on the Si_3_N_4_/Si/Si_3_N_4_ substrate when subjected to concentrated HCl. (**c**) The blank control group containing the culture medium without HeLa cells also shows no obvious temperature drop after the injection of concentrated HCl. (**d**) Temperature changes of the culture medium caused by the exothermic effects of concentrated H_2_SO_4_ and of concentrated HCl. (**e**) The morphology of HeLa cells at the moments 1 min, 2 min, and 5 min after the introduction of concentrated HCl.

## Data Availability

The data presented in this study are available on request from the corresponding author.

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
