# Peer review of "Non-Interventional and High-Precision Temperature Measurement Biochips for Long-Term Monitoring the Temperature Fluctuations of Individual Cells"

_biosensors, 2021, doi:10.3390/bios11110454_

Round 1

Reviewer 1 Report

The proposed manuscript within the scope of the Biosensor journal. Some of the results observations are interesting, however, if the paper can be improved in the following areas, it would add more value to the readers:

  1. Please illustrate the variation of parameters of different lines in Fig. 3(a), such 1Au2, 1Bu3 and…. Etc.
  2. The Figure 3 (d) shows the thermal noise of the commercial thermocouples, but the comparison between the developed and commercial sensors and the advantages of the developed sensors should be added.
  3. How to determine the accuracy and repeatability of the developed sensor, and please described the repeatability performance of this sensor?
  4. Whether the position of cell affect the detecting temperature of this sensor?
  5. How about the measuring accuracy of the sensor at each temperature range?
  6. Please illustrate how to calibrate and compensate the sensing variation that generated by sensor fabrication process.

Reviewer 2 Report

Temperature detection for a single cell is very important both for biological science and sensing technology. I think this manuscript contains some interesting things to this area. The data are clear and the device seems valuable. So I recommend acception after a minor revision. The key point lies in Figure 5. Why did the authors use HCl to induce the death of cells? The reaction of HCl with buffer is complex. I don't think the data is difficult for analysis. I suggest the authors should use some drugs or biological small molecules to change the cellular temperature. Then the reason for temperature alteration should be discussed.  
